# Homo Interneticus—The Sociological Reality of Mobile Online Being

**Bogdan NADOLU \* and Delia NADOLU**

Department of Sociology, West University of Timisoara, 300223 Timișoara, Romania; delia.nadolu@e-uvt.ro
\* Correspondence: bogdan.nadolu@e-uvt.ro; Tel.: +40-723235371

**Abstract:** The digitalization of everyday life has become a common place reality for more than half of the global population. Being connected 24/7 on several devices, being only one click/touch away from a huge amount of digital content, being available for interactions with almost any users from around the globe have become routine. In this paper, we identify the main sociological dimensions of the so-called Homo interneticus—a new manifestation of the human condition—on the basis of new communication technologies. The main research question was: Is time spent online a valid predictor of smartphone addiction? We conducted an experimental survey using a standard online questionnaire incorporating the Smartphone Addiction Scale—Short Version, followed by a request to upload screen captures recording respondents' phone use during the previous week. We gathered 140 responses between May and July 2019. These were anonymously analyzed. The consistency between self-estimates and phone logs that we found gives encouragement for the development of this approach. A main finding is that the daily time spent on smartphone use is not in itself a strong predictor of addiction. Our research suggests that future studies should distinguish types of usage and investigate motivational springs.

**Keywords:** internet; computer-mediated social interactions; social impact of digital technology; Homo interneticus

## 1. Introduction

In the last three decades, human civilization has experienced an unprecedented development of communication technology, with direct and complex consequences for all dimensions of our lives [1]. The generalization of the Internet and the New Information and Communication Technologies (NICT) represents a global social phenomenon that has occurred largely from the 1990s on, with a totally unpredictable trajectory and in an uncontrolled, quasi-anarchic way. Starting in 1969 as a military research project conducted by ARPA (Advanced Research Projects Agency financed by the US Government), the global network of computers (named firstly ARPANET and, after the 1990s, the Internet) has generated a new stage in our evolution: the information society [2]. Its original design purpose—to enable survival during an atomic war by developing a robust communication network—has resulted in today's huge digital universe of computers, servers, satellites, digital resources and, of course, users who in fact do not belong to any organization. On the Internet, anyone can say anything, anytime, from anywhere [3]. If any software or application tries to restrict access to its contents, there are all the time other similar ones that can be used without any problem. Unlimited access to almost the entire digital content is probably the first premise of the development of the Internet.

Another important aspect is related to the development of technology. While in the 1980s the personal computer was a rather prohibitive tool, both in price and in requisite skills, at the present day, due to technical developments, digital technology is very affordable, and more importantly, its use

is strongly focused on non-specialist domestic end-users. Every six months, new, more powerful smartphones are launched onto the market, capable of doing things unimaginable only a couple of years before. With every launch, the price of previous versions goes down, so that a 2015, $1000 mobile phone could be bought for less than $200 in 2019. Access to cheap, friendly technology represents the second premise of the development of the Internet.

One of the most important facilitators of the rapid spread of accessing the Internet can be associated with its hedonistic dimension. In addition to its general value as a real and complex tool for much of our daily activity, we have to admit that digital technology has also proved to be a potentially unlimited resource for leisure purposes: movies, music, pictures, games, chats, and so on. During the 1990s, one of the most frequent points of discussion regarding the negative impact of the Internet was pornography. Today, the impact of online computer games played by teenagers all over the world has become another important concern [4–6].

In this context, we access the digital universe daily because we face no restrictions to going online, we can afford suitable technology, and, last but not least, we enjoy exploring this new world. The digitalization of everyday life has become a common reality for more than half of the global population. Being connected 24/7 on several devices, being only one click/touch away from a huge amount of digital content, being available for interactions with almost any users from around the globe, all these possibilities have become commonplace. The penetration of communication technology into our daily life is both more extensive and more intensive than ever before. Our values, attitudes, behaviors, expectations, sociability, daily routine, and social life have been reconfigured by the use of digital technology [7].

In 2004, Michael Goldhaber coined the expression Homo interneticus (HI) to describe a "new form of human evolution based on [humans'] state of communication abilities". Although, at first sight, this may seem little more than a catchphrase, we should recognize that it is tending to become more and more legitimate. The effect of the Internet is contributing to human evolution: " . . . the invention of language was an evolutionary step of the humankind. The internet is a similar step" [8]. Another reference to this formulation can be found in Aleks Krotoski's documentary movie "Homo Interneticus?—The Virtual Revolution" screened by BBC2 in 2012 [9]. The main theme of this film was how the web may be distracting and overloading our brains. Another more widely used term to capture this reality is Homo Digitalis (HD) [10,11]. Even though these are almost synonymous, HI can be considered as somehow comprehending HD. Someone with several digital gadgets but no Internet access is HD but not HI. It is also true that during the early history of the spread of the Internet, much of the communication technology depended on existing analogue devices, so HI could be considered the more valid original descriptor, while HD is arguably a more novel one.

A Google search for "Homo Interneticus" will yield many results (15,700 compared with 377,000 for HD in December 2019), albeit dwarfed when compared to a search for the term "internet" (7,360,000,000 results). Probably, the best explanation for the relatively restricted use of this term is that it is related to a quite common reality. We are still accessing the Internet and using digital technology even if we are not constantly self-consciously labelling ourselves. The problem is that the difference between digital people, native or immigrant [12], and non-digital people is becoming more and more complex. At the lowest level, these differences are related to a redefinition of the basic social skills that involve us in community issues, civic participation, alienation, neighborhood relationships, and other unmediated interactions in the public space. At the intermediate level, they are about cognitive resources and access to the infosphere, as well as connection with mainstream information fluxes and access to global virtual sociability [13–17]. It has even been claimed that epigenetic mechanisms may come into play here, leading to an atrophying of our powers of memory and a reduction of our attention span; we are also negatively impacted by the demands of engagement in multitasking, icon-based communication, and extensive possibilities for psychological disturbance, with Internet Addiction Disorder (IAD) becoming a recognized mental problem. All these consequences of the intensive use of

digital technology reinforce the need for the scientific investigation of what happens to Internet users following such a massive insertion of digital technology into their lives [18–21].

Every one of us who uses digital technology for at least a few hours every day, without any pause, sooner or later becomes a Homo interneticus. It does not matter what we call ourselves; it only matters that we are using these technologies intensively, and the effects of that use are unavoidable. We seek neither to fight against this trend nor to promote it; it is simply that a global phenomenon which affects more than half of the world's population [22] provides a legitimate and substantial theme for sociological investigation.

The main characteristic of Homo interneticus is his/her online presence. If, in non-mediated reality, our existence is implicitly defined by our social life (and we exist in social space as long as we are alive), in virtual space our existence is strictly conditioned by our communicative action. In other words, if on the Internet we do not engage either as receivers or as senders, then to all intents and purposes we have no existence there. If we do not read or send any email, post on social media, or publish any audio–video content, then in practice we do not exist in virtual space. It is completely pointless to have an e-mail account that we never access, to have a Facebook account that we never read or post to, to have any kind of online enrolments and not to access them in any way. Thus, one of the most important dimensions of Homo interneticus is time spent online, and this represents the central idea of this paper.

We started out from a classic question about Internet use: How many people use the Internet, and how much time do they spend online? If the precise number of Internet users is more of a rhetorical question (due to the dynamics of the extension of the Internet and, more importantly, due to difficulties in defining the term 'user'), time spent online requires a quite different approach. Today, it is very easy to have 24/7 access to the Internet from smartphones. The answer to the classic question "How long did you spend online?" is no longer to be calculated in hours. Usually people go online for as long as they need and whenever they feel like it. Besides this reshaping of online time, there are also questions about how long we spend online daily or, in other words, how many hours we are not offline.

## 2. Materials and Methods

Starting from this issue of accessing the Internet, we developed our main research question: *Is time spent online a valid predictor of mobile addiction?* For this, we conducted an experimental survey using a standard online questionnaire (five items) together with the Smartphone Addiction Scale—Short Version (SAS—SV) [23,24] and followed up with a request to participants to upload screen captures of phone usage during the previous week (Appendix A). This last part of the questionnaire was a real challenge, due to difficulties respondents faced in obtaining the technical information from their smartphones. Thus, those with iPhones only had to access Settings and make two screen captures with the time of screen use and the most frequently used applications. We asked non-iPhone to install a small free application, UsageTime [25], which could immediately provide data on their phone utilization for the last week. We included in the online questionnaire a very detailed explanation of this procedure and obtained 140 responses during the period May–July 2019. From a methodological point of view, we used a purposive sampling method focusing on experienced users. This was, therefore, only a convenience sample, with subjects approached via Facebook using the snowball selection technique. We acknowledge that we cannot therefore claim full representativeness but consider that the results can provide a useful point of departure for further work. The subjects were informed about the exclusively research purpose of the study and its anonymous and confidential character. The statistical analysis was chiefly focused on the profile of the four types of HI and on a comparative analysis of the degree of consistency between the subjective self-estimation of online behavior and the objective data recorded by the smartphones of the respondents. The entire research design, including the questionnaire and the research report, was approved by the Ethics Committee of West University of Timisoara, Romania (2625/0–1/20.01.2020 RCE2020–13).

## 3. Results

From a descriptive perspective, the answers recorded for factual items showed the following distributions (Figure 1):

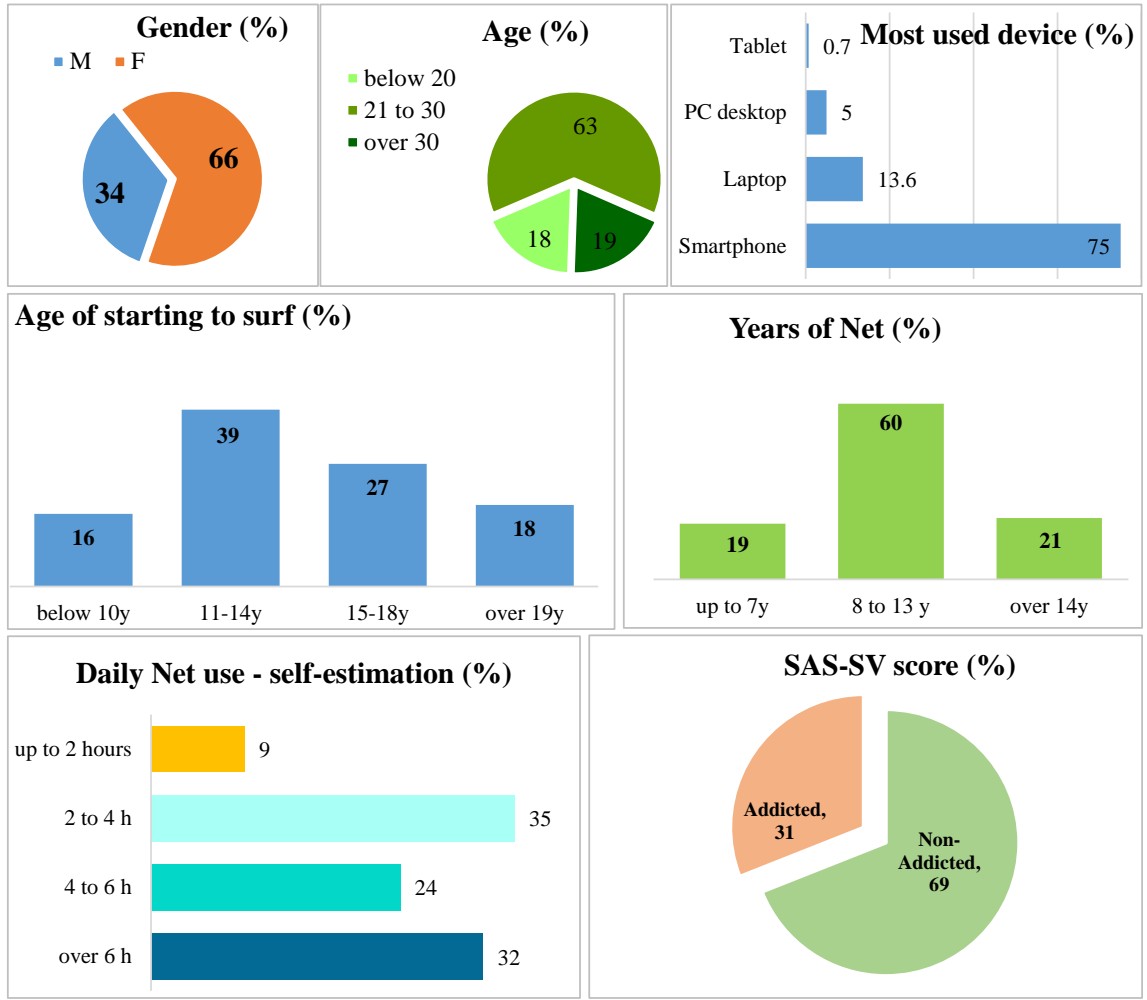

**Figure 1.** Descriptive statistics for the factual distributions and Smartphone Addiction Scale—Short Version (SAS–SV) scores.

The central theme of the questionnaire was related to the amount of time spent online. This was quantified in two ways: through self-evaluation of daily online time from any device (Q2) and by means of a screen capture of the mobile phone being used (the second section of the questionnaire). On the basis of the self-evaluation of daily Net use, we defined four categories of users: Non-Homo Interneticus (Non HI)—up to 2 h of use, Light Homo Interneticus (Light HI)—2 to 4 h of use, Medium Homo Interneticus (Medium HI)—4 to 6 h of use, and Strong Homo Interneticus (Strong HI)—over 6 h of use. Our chosen cut-off point of 2 h between Non-HI and HI arose from a value determined by a weighted statistical analysis of public domain information (such dailyinfographic.com, businessinsider.com, or statista.com) about the volume and partitioning of online activity (Google searches, e-mails, Facebook, Twitter, YouTube watching, and so on) multiplied by an estimated amount of time and divided by the known number of users for each application. Even if this approximated number may seem quite large, it was based on publicly available statistics concerning Internet activity. The 2-h figure was established by dividing the global internet use of all kinds by the number of users. Of course, some people spend less than this mean time online daily, while others spend more. We decided, for the present purposes, to use this threshold of 2 h of daily online activity to demarcate between non-HI and HI. HI is thus our

shorthand for "someone with above-mean usage of the Internet". The main characteristics of each type of Homo interneticus are presented in Table 1:

**Table 1.** Frequency of Internet use and characteristics of various types of Homo interneticus (HI) in our study sample.

|  | % | Average Age | Average Net Start | Average Mobile Use | Average Peak Mobile Use |
|---|---|---|---|---|---|
| up to 2 h (Non-HI) | 9 | 38 y 9 m | 22 y 1 m | 2 h:14 min | 3 h:21 min |
| 2 to 4 h (Light HI) | 35 | 27 y 1 m | 16 y 3 m | 4 h:15 min | 6 h:22 min |
| 4 to 6 h (Medium HI) | 24 | 25 y 2 m | 14 y 11 m | 4 h:44 min | 7 h:20 min |
| over 6 h (Strong HI) | 31 | 25 y 1 m | 13 y 11 m | 4 h:39 min | 7 h:16 min |

The analysis of these data yielded a significant negative correlation (r = −0.237, $p < 0.007$) between Internet use and age: users who access the Internet for less than two hours daily (Non-HI) have an average age of almost 39, started to use the Internet at around the age of 22, use their mobile phones on average for 2 and a quarter hours daily, and the average of their peak mobile usage (the highest values during the last week, as recorded by screen capture) is 3 h 21 min. All these variables show a direct inverse linear correlation with age. If we compare the two extremes (Non-HI vs. Strong HI) we see that users who spent more than 6 h daily online (Strong HI) have an average age of around 25, started using the net at 13, and use a mobile phone for more than 4 and a half hours daily. Their peak mobile use excess compared with their average mobile use is the highest of all groups shown (with their average peak of mobile use in a week being 7 and a quarter hours).

The experimental part of the survey made use of the uploading by each user of a monitoring app for smartphone utilization over the previous week (calculated from the day when they started to fill in the questionnaire). Even though participants experienced some difficulties in fulfilling this request (given the variety of types of operating systems for mobile phones), subjects did manage to send screen captures with these data in a completely anonymous way. The statistics of mobile phone utilization (as automatically recorded by each device) are presented in Table 2, and the type of activity in Figure 2:

**Table 2.** Mobile phone use (self-recorded data).

|  | % |
|---|---|
| up to 2 h (Non-HI) | 15 |
| 2 to 4 h (Light HI) | 30 |
| 4 to 6 h (Medium HI) | 38 |
| over 6 h (Strong HI) | 17 |

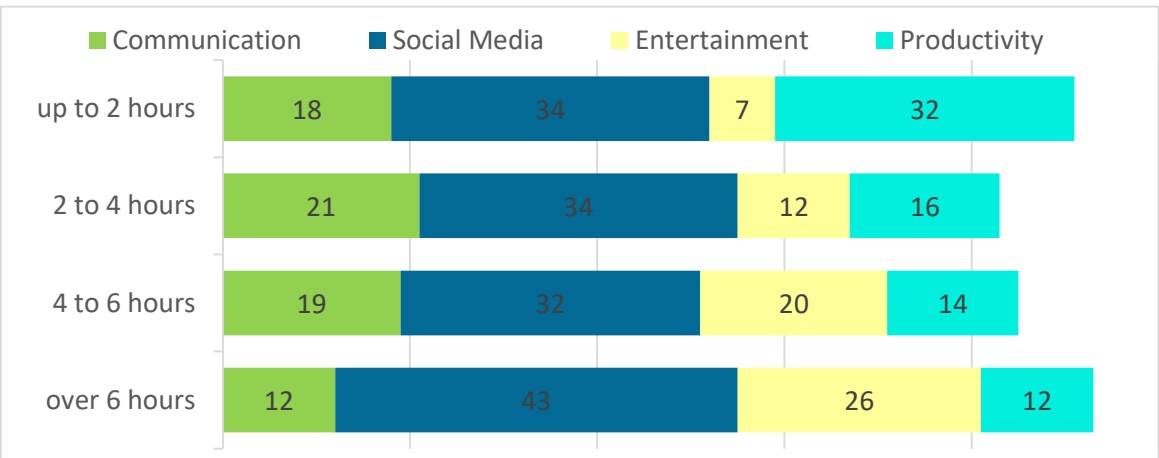

**Figure 2.** Types of activity recorded by phones (%).

Directly related to daily mobile phone use, we included in the questionnaire an item about self-evaluated frequencies for this usage in various contexts and its results are presented in Figure 3:

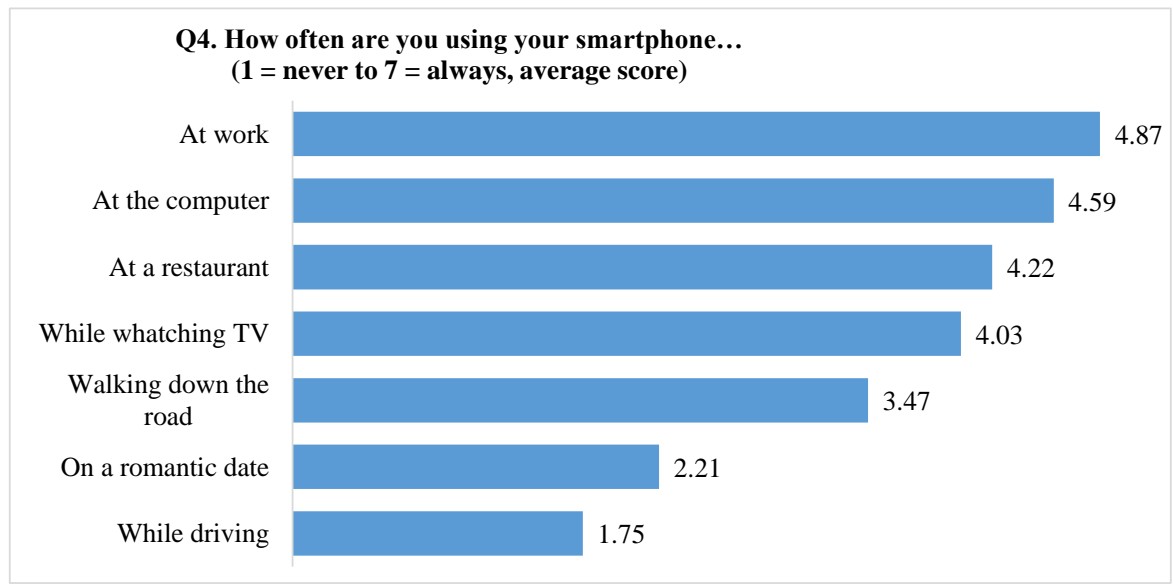

**Figure 3.** Average self-estimated values for smartphone usage in various contexts.

Observing this distribution pattern, we can distinguish four contexts in which mobile phone use frequency is above average: work, computer, restaurant, and TV. In all of these, the use of a smartphone is understandable.

A specific aspect explored by the questionnaire concerned the digital disconnect experience, as a previous fact (Q3) or as a possibility (Q5). Even though the hypothetical prospect of discontinuing phone use involved only a personal estimation, for these questions, more than two-thirds of subjects both reported having experienced digital disconnect and claimed that they could contemplate it. (Table 3):

**Table 3.** The digital disconnect experience.

|  | YES | NO | Less than 1 Day | up to 3 Days | 4–7 days | More than 7 Days |
|---|---|---|---|---|---|---|
| Q3. In the last 6 months, did you spend a certain amount of time without access to the Internet? If yes, how long have you been offline? | 75 | 25 | 12 | 13 | 11 | 8 * |
| Q5. Could you temporarily give up access to the Internet from your mobile phone? | 91 | 7 | 6 | 29 | 14 * | |

\* Discrepancies between totals and 100% are accounted for by non-responses.

Neither the disconnecting experience nor disconnecting predictions appeared statistically correlated with HI profiles: $F = 0.735$, $sig < 0.393$ for previous disconnection experience and $F = 0.179$, $sig < 0.674$ for willingness to contemplate temporary disconnection.

Finally, an investigation of a separate topic not addressed in the questionnaire—that of smartphone addiction—was operationalized by applying the SAS—SV, using a version re-calibrated for Romania.

The cut-off values were 31 for males and 33 for females (anything below these values signifies no addiction). For the entire sample, 69% of subjects obtained a non-addicted score. The Cronbach Alpha coefficient was 0.871. The SAS–SV scores cross-tabulated with the factual data are presented in Tables 4 and 5:

**Table 4.** SAS–SV scores cross-tabulated with factual data.

|  | % | Male | Female | below 20 | 20–30 | over 30 |
|---|---|---|---|---|---|---|
| Non-addicted to smartphone | 69 | 76 | 65 | 60 | 63 | 68 |
| Addicted to smartphone | 31 | 24 | 35 | 40 | 38 | 32 |

**Table 5.** SAS-SV average scores for factual variables.

| Gender | SAS–SV Score | Age | SAS–SV Score | Daily Mobile Phone Use | SAS–SV Score |
|---|---|---|---|---|---|
| Male | 25 | below 20 years | 32 | up to 2 h | 21 |
| Female | 30 | 20–30 years | 28 | 2–4 h | 27 |
|  |  | over 30 years | 25 | 4–6 h | 31 |
|  |  |  |  | over 6 h | 32 |

Even though the female average SAS–SV score was higher than the male average, the gender difference was not significant (Pearson Chi-Square = 1.607, sig = 0.205). There were significant differences with respect to user age (F = 4.062, df = 2, sig = 0.020) and daily mobile phone utilization (F = 3.517, df = 3, sig = 0.18).

In Figure 2, we can see how the proportion of entertainment activity increased with increasing smartphone utilization time, along with the decreasing use for productive purposes. Time spent on the smartphone showed a significant positive correlation with the SAS–SV scores (r = 0.321, $p < 0.001$) and with mobile entertainment activities (r = 0.267, $p < 0.01$) and a significant negative correlation with mobile productive activities (r = −0.263, $p < 0.01$).

The causality between time spent on the mobile phone as an independent variable and the SAS–SV score (as a dependent variable) was evidenced by the linear regression model (Beta = 0.321, t = 3.391, sig < 0.001). The score for SAS–V was also subjected to multiple regression analysis, using total time spent on the mobile phone and the proportions of time devoted to each kind of activity as predictors. When this was done, the relative amount of time spent on social media activities emerged as the most important predictor (importance = 0.57, $p < 0.001$).

## 4. Discussion

Our starting point was the research question: *Is time spent online a predictor of smartphone addiction?* Our expectation was that, due to some specific professional activities, some users have to spend a significant amount of time online, which does not necessarily imply a state of addiction. In other words, addiction needs to be interpreted in terms of the type of online activity and not simply of the amount of time spent.

We might broadly categorize online activity as production-orientated (done primarily from motives of practical necessity) or hedonistic (done for the sake of pleasurable reward). Either of these might have compensatory or addictive dimensions. The concept of compensatory behavior (undertaken in place of resolving or facing up to problems in non-mediated social reality) is a well-discussed and defined phenomenon [26]). A workaholic would be a compulsive production-orientated agent. A social loner might seek compensatory affirmation through online relationships without presenting characteristics of addiction. Hedonistic internet use might reflect any vectorial combination of intensity on the compensatory or addictive axes. Clearly, in such a complex situation we cannot expect to isolate the variables and predictors at all easily.

We cannot prove from this study that either compensatory or compulsive (that is, addictive) factors explain the observed inflation of social media and entertainment usage in our Strong HI group, which is evident from the pattern shown in Figure 2. However, the trend is sufficiently suggestive to encourage us to investigate the possible interactions more precisely in future studies. It would,

for example, be interesting to harvest more precise data on internet usage during an immediate rebound from periods of internet deprivation, particularly if it were possible to distinguish voluntary and involuntary internet disconnect episodes.

From a sociological perspective, these findings can be interpreted within two paradigms—those of functionalism and symbolic interactions. Computer-mediated communication via the Internet is probably the best representation of the functionalist model. Each user can choose what to do in virtual space, what to read, see, follow or interact with on the basis of personal goals, expectations, values, and preferences. The features of user-generated content and unrestricted possibility of interaction in virtual space fit well with the Utilization and Gratification Model [27]. Supplementing this functionalist perspective, we may remark that by coming online from all over the world, with different cultural, social, and economic baggage, Internet users engage in symbolic interactions based on their personal experiences, which become temporarily defined in correlation with their purposes [28]. In this context, the extended amount of time spent online by a native or immigrant digital should perhaps be seen as largely a function of the present condition of our lives, rather than being characterized as a primary sign of addiction.

## 5. Conclusions

We labelled as Homo interneticus Internet users who spend more than two hours daily online using various devices, but chiefly smartphones. Smartphone addiction is directly related to the amount of time spent on this device but is also related to the type of activity, with the most important predictor being the pattern of accessing social media. One of the main characteristics of use of these platforms is the practice of pseudo-infinite scrolling. In other words, the constant updating of the content captures the attention of the users and keeps them focused on that application, which thus locks them into a never-ending story [29,30]. A complementary dimension of these platforms is their hedonistic effect, and thus, the amount of time spent online scrolling can become substantial. In our sample, the mean daily time spent, as recorded by the smartphones themselves, was 6 h 40 min, with the maximum cumulative time spent on a mobile phone by one person in a single day being 18 h.

Our most important observation is that, proportionally, Strong Homo Interneticus cybertime allocation prioritizes social media access and entertainment at the relative expense of the productive and communicative functions that we would intuitively expect to be less associated with hedonistic, compensatory, or addictive use. It remains to be established, however, whether this apportioning of online activity genuinely reflects underlying dynamics of intrinsic or extrinsic motivation.

A second, and perhaps surprising, finding is the degree of concordance between self-estimated internet use and the objectively recorded data from the smartphones accessed through retrospective screen captures which the participants had not previously known were to be requested. This gives us confidence in the usefulness of such a component in future research work, which is important given the need, especially in studying this field, to relate to the population being investigated using techniques that are congenial to them.

The main limitation of this study has to do with the sample profile of the respondents with significant online activity (mostly young, educated people, largely female). Another limitation is the lack of easy comparability between the self-recorded screen captures submitted by each phone user (since different phone operating systems generate different summaries of the data, with different levels of visual resolution, which limited the precision of our data extraction). In future research, an approach using field trained operators might improve this aspect.

The near future will show whether all this digital empowerment represents genuine progress or is merely a fashionable trend [31].

**Author Contributions:** Conceptualization, B.N. and D.N.; methodology, B.N. and D.N.; validation, B.N. and D.N.; formal analysis, D.N.; investigation, B.N.; data processing, D.N.; writing—original draft preparation, D.N.; writing—review and editing, B.N. and D.N. All authors have read and agreed to the published version of the manuscript.

**Funding:** This research received no external funding.

**Acknowledgments:** Special thanks to Sorin Matei of Purdue University for very useful feedback and reference recommendations and to Kevin Cox of Ohio State University for friendly feedback concerning the interpretation and for very useful proofreading in English. The authors gratefully acknowledge the contributions of Dorothy and Stuart Elford, of Timișoara, in helping to refine the English expression and presentation of the content and for their academic criticism, which has improved the clarity of the argument.

**Conflicts of Interest:** The authors declare no conflicts of interest.

## Appendix A

Anonymous Questionnaire—Using the Smartphone (English version).

*Hello! We are conducting a piece of experimental sociological research on the use of smartphones. For this, please answer the questions in the following questionnaire and, as far as possible, help us by measuring the time of use of your phone. The data you provide will be integrated into a global analysis, without your being identified. We guarantee your anonymity and confidentiality. By completing this questionnaire, you accept the use of your answers in an aggregated statistical way. Thank you!*

Question 1 (Q1). From what age have you constantly accessed the Internet?
Q2. On average, how many hours do you spend daily on the Internet? (from any device)
Q3. In the last 6 months, did you spend a certain amount of time without access to the Internet? If yes, how long were you offline?
Q4. To what extent do you access the Internet from your mobile phone in the following situations (see Table A1):

**Table A1.** Situations when it is accessed the Internet from the mobile phone.

| | Never | Very Rarely | Rarely | Moderately | Often | Very Often | Always |
|---|---|---|---|---|---|---|---|
| walking along the road | | | | | | | |
| at the wheel | | | | | | | |
| at outdoor cafes/restaurants | | | | | | | |
| at work/school | | | | | | | |
| while using a computer | | | | | | | |
| while watching a TV show | | | | | | | |
| during dating/romantic moments | | | | | | | |

Q5. Could you temporarily give up access to the Internet from your mobile phone? If yes, for how long?
Q6. Please evaluate the following statements (see Table A2):

**Table A2.** Smartphone Addiction Scale—Short Version.

|  | Strongly Disagree | Disagree | Partially Disagree | Partially Agree | Agree | Strongly Agree |
|---|---|---|---|---|---|---|
| 1. It happens that I miss tasks and planned activities due to my use of the smartphone |  |  |  |  |  |  |
| 2. I have moments when I have to focus hard on what I am doing, because of using a smartphone |  |  |  |  |  |  |
| 3. I feel pain in my wrists or throat while using my smartphone. |  |  |  |  |  |  |
| 4. I can't be without a smartphone. |  |  |  |  |  |  |
| 5. I feel restless and agitated when I don't have the smartphone at my fingertips. |  |  |  |  |  |  |
| 6. I think about the smartphone even when not using it. |  |  |  |  |  |  |
| 7. I will never give up using my smartphone. |  |  |  |  |  |  |
| 8. I constantly check the smartphone to see what news has appeared. |  |  |  |  |  |  |
| 9. I use my smartphone more than I intend. |  |  |  |  |  |  |
| 10. People around me tell me I use my smartphone too much. |  |  |  |  |  |  |

**Time of use of your smartphone**

Q7a. If using an iPhone type, please go to settings/battery (settings/battery) or settings/screen time (settings/screentime) and make TWO SCREEN CAPS (center button + off button or off button + volume up), one recording the use of the phone, and one of the most used applications IN THE LAST WEEK.
Q7b. If you use another type of phone, please install the free Usage Time application (size 4Mb) and make TWO SCREEN CAPS, one of phone usage, and the other of the most used applications IN THE LAST WEEK.
Q7c. If you are unable to load the screenshots, please fill in your data in the table below (See Table A3):

**Table A3.** Number of hours per day.

| Number of Hours |  | Number of Hours |  |
|---|---|---|---|
| 1st day |  | 5th day |  |
| 2nd day |  | 6th day |  |
| 3rd day |  | 7th day |  |
| 4th day |  |  |  |

Q7d. If you have not been able to load the screenshot, please specify the most used applications, in the order in which they appear (the first 10)
Q8. For statistical processing please specify:
GENDER M F
Age: _______
You most often access the Internet from ... Answer: _________________
How many virtual friends do you have? (approximately) Answer: ______

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
