# Peer review of "Homo Interneticus—The Sociological Reality of Mobile Online Being"

_sustainability, doi:10.3390/su12051800_

Round 1

Reviewer 1 Report

This paper is well planned and interesting. But I am recommending revision, resubmission, and reconsideration for the following reasons, presented in descending order of significance:

By your own admission, at lines 119-121, the entire study was based on a convenience sample, conducted using "snowball" techniques. Early respondents with a strong motivation therefore played a role in recruiting other respondents. As you concede, this methodology defeats any claim to representativeness or scientific validity.

There may be subjects of study so compelling and so novel that we can forgive convenience sampling in the exploratory stages. This paper is neither of those things. That doesn't destroy its merit. It just means that you have to do the grueling work of collecting good, scientifically reliable data in the first instance.

You devote an entire page to an attempt to establish Homo interneticus as a clever neologism. I did not find this particularly successful or useful. You have found some interesting trends in Internet use. But as your own Google search of the word Internet establishes, web browsing is a ubiquitous, perhaps universal trait of people with the financial means to indulge this habit.

As Romanians, you are true modern guardians of the linguistic legacy of Latin! The trouble is that net, the second half of Internet, is a purely Germanic word with no clear relatives in Latin or the Romance languages. https://www.etymonline.com/word/net. Taxonomic names in biology sound best when they stay with Latin and Greek.

As an author who has used the term Homo economicus, I do understand what you're trying to accomplish. But you have good points of sociology within your reach. I think that you weaken rather than strengthen your social science with this attempt at rhetoric. I recommend that you nail down the science first. Then you can revisit your effort to find the right name for this sort of human being. Ecce homo!

I am not at all certain how this article, regardless of its merits, fits in the broader mission of Sustainability. There are other MDPI journals better suited to your topic. I would recommend that you work with your contact at Sustainability to find a better placement within this family of journals.

If you can duplicate your results with a scientifically reliable sample, I would be much more inclined to recommend publication. As the paper stands, I believe that reliance on a convenience sample poses a fairly substantial barrier to receiving positive peer review. It might not be fatal; I am sure others could and would disagree. I am impressed by your command of sociology in every other respect, however, and imagine that it would not detain you long or cause you undue inconvenience for you to sharpen your surveying techniques.

I enjoyed the paper. I wish you the best of luck with it.

Reviewer 2 Report

An extremely interesting and well researched paper. Very good, if short, conclusion. Very clear description of methods and results.

Excellent and accurate set of references, some of which I need to track down and read such as Nadin’s book. Only one typo I spotted in item 21 where “versiono f” should be “version of”.

This is a fertile research field which will reward further work. While this is primarily about smartphone usage (as pointed out at lines 228-229, I wonder whether interneticus-style behaviours differ across the less portable platforms such as tablets and PCs – for example for those using multiple platforms, do the behaviours vary? Another topic which I would find interesting relates to the type of content being accessed – there is a little in there, but it would be interesting for further research to understand the languages of the material being accessed and whether this varies across the sample (in the Romanian case, is it mostly Romanian, French, English, German, Italian or Russian, for example?). I am interested in this from the perspective of colonisation by language in relation to cultural change and outcomes – and the mode of accessing technology affects this process.

Line 44: earlier models are always cheaper simply because they are regarded as “old” technology and harder to sell, and certainly the point is valid. However, new models of smartphones are always much more expensive than the initial price of the previous model. While this paper is about usage and addictive behaviours, I just wonder whether the lust for the latest expensive model is in itself a symptom of addiction. There was not a question on this in this survey, but might be a thought for any repeat of the survey – repeating the survey may also identify any changes over time.

Table 3 is very interesting. Was any analysis done of non-response. It appears that for some questions non-response was quite high and may have introduced biases. (just a side note: “non-response” is more usual than non-answers)

Line 214 has a superfluous “e” in the middle of the line.

Appendix A: Translation is a bit stilted in places although the meaning is clear. Also some errors in the labels in the tables.

Important points: (1) “below” is repeatedly typed as “bellow” [fig 1 and tables 4&5], (2) “productivity” in fig 2 is missing a “t”, (3) preamble at line 258 would be better English as “By completing this questionnaire you accept the use of your answers in an aggregate statistical way”, (4) line 260 Q1 “From what age did you constantly access the internet” (5) question 6 would be better as “I think about my smartphone when not using it” or if you have an intensitive in the original “ … even when not using it” and (6) question 10 in the appendix should be “People around me tell me I use my smartphone too much”

Reviewer 3 Report

This study fleshes out some key findings and identifies challenges pertinent to contemporary effects of technology on everyday life. The contribution of this paper is also its strongest point. The paper, nonetheless, requires some amendments to strengthen its quality.

Methodologically, what were the ethical considerations (especially regarding the sharing of screenshots, as well as the possibility that minors might have taken part)? Was there ethical clearance by an Ethics Committee? In the methodology section, the authors need to discuss further their method of analysis of the data. In the findings, some key correlations are highlighted, yet the discussion section is underdeveloped and indeed contains information that one might expect to read in the results. The discussion section needs proper attention and enlargement to explore the findings in relation to other knowledge available in the field. For example, how do the findings of this study relate or situate in relation to Homo Interneticus? Or, what about applying social theory of internet use to discuss the findings from this study? This section (i.e. discussion) needs to emphasise on the novel contribution of this paper/study. A section that discusses the limitations of the study is important. When contextualising the study (initial section of the paper), have the authors thought of considering the social functions of the use of the internet (functionalist perspectives) in order to help better understand the findings regarding age and internet use?

Round 2

Reviewer 1 Report

Thank you for your extensive and thoughtful response to my initial review. Let me address and resolve the least substantial of my original critiques:

  1. Homo interneticus versus Homo digitalis: This was by no means a barrier to publication! I simply wanted to point out my own sense that interneticus lacked both music and history. Music, because it's a five-syllable word that forces the natural rhythm of its root (In'-ter-net") to shift (In"-ter-net'-i-cus) because few if any words in any language can bear primary stress on the first syllable where there are more than three syllables. History, because it remains an awkward and forced transformation of an English word back into Latin.

    I do appreciate the subtle distinction relative to Homo digitalis. The word digit does more work than you might appreciate: In addition to referring to numbers, digit also speaks of the fingers. That's the instrument humans use to navigate their phones! (At least for now.)

    Latin words emphasizing connectedness, as opposed to mere electronic devices not linked to the World Wide Web (three consecutive Germanic words!), include iunctus and conexus. The Germanic word net does come fairly close to the Latin root in words such as connect and nexus.

    Finally, you do find some instances of Homo interneticus, though they are few, recent, and far less prevalent than other terms. Still, you do an admirable job of defending your neologism. Consider yourself vindicated.

  2. If the editors of Sustainability are happy to include this article in their journal, I'll retract my sense of dislocation. My initial connection with this journal arose through the more common denotation of sustainability, which emphasizes exhaustible and renewable natural resources. The idea of social sustainability, apart from this physical sense, remains controversial among environmental economists.

    If you care to add a single paragraph on that point, I would enjoy seeing how you handle this issue. By no means would I compel you to do this. And I will warn you that the literature contesting this issue is both violent and deep. As someone who favors a purely physical definition of sustainability and would prefer finding another term to describe social, economic, and political stability, I did not instinctively embrace your study as an expression of sustainability.

    Ultimately this is a question for the editors. If they are comfortable with your article in their journal, I have no reason to contest their judgment.

  3. All right. That brings us to the true point of disagreement in my original review. I appreciate the daunting nature of conducting a truly representative sample. Efforts to manage the scale of such a project, such as limiting the inquiry to highly developed countries or offering a tangible incentive to survey participants, are likely to be prohibitively expensive.

    I can live with the idea that you have crafted a useful purposive sample, even if remains one from which one cannot extract statistically meaningful generalizations to the broader population. Your explanation is clear and helpful, even if I still have some misgivings. As I said in my first review, I believe that there are instances where exploratory studies based on non-representative sampling warrant publication. You've gone a long way toward defending this project as one of those instances.

All that remains is whether to recommend publication outright or to urge further minor revisions. I see no reason to delay a positive recommendation.

I do hope you consider breaking your prose into more paragraphs. In particular, the new material you have added in response to my first review, especially at lines 73-78, 89-93, and 134-143, lengthen your original paragraphs past comfortable limits of legibility and coherence. Please consider setting this material into additional paragraphs.

After all, one of the truly attractive features of this article has been its cadence. You tell a good story. Don't let your valiant and valid efforts to a critic ruin the rhythm of the tale you tell.

Good luck with this paper and your future research. I enjoyed reviewing this article and interacting with you.

Reviewer 3 Report

Thank you for this revised version of your paper. The amendments have indeed improved its overall quality.